# Local magnetic moments in iron and nickel at ambient and Earth's core conditions

A. Hausoel[1], M. Karolak[1], E. Şaşıoğlu[2,3], A. Lichtenstein[4], K. Held[5], A. Katanin[6,7], A. Toschi[5] & G. Sangiovanni[1]

Some Bravais lattices have a particular geometry that can slow down the motion of Bloch electrons by pre-localization due to the band-structure properties. Another known source of electronic localization in solids is the Coulomb repulsion in partially filled $d$ or $f$ orbitals, which leads to the formation of local magnetic moments. The combination of these two effects is usually considered of little relevance to strongly correlated materials. Here we show that it represents, instead, the underlying physical mechanism in two of the most important ferromagnets: nickel and iron. In nickel, the van Hove singularity has an unexpected impact on the magnetism. As a result, the electron–electron scattering rate is linear in temperature, in violation of the conventional Landau theory of metals. This is true even at Earth's core pressures, at which iron is instead a good Fermi liquid. The importance of nickel in models of geomagnetism may have therefore to be reconsidered.

[1] Institut für Theoretische Physik und Astrophysik, Universität Würzburg, Am Hubland, D-97074 Würzburg, Germany. [2] Peter Grünberg Institut and Institute for Advanced Simulation, Forschungszentrum Jülich and JARA, 52425 Jülich, Germany. [3] Institut für Physik, Martin-Luther-Universität Halle-Wittenberg, 06120 Halle (Saale), Germany. [4] Institut für Theoretische Physik, Universität Hamburg, Jungiusstrasse 9, 20355 Hamburg, Germany. [5] Institute of Solid State Physics, TU Wien, 1040 Vienna, Austria. [6] M. N. Mikheev Institute of Metal Physics, 620990 Ekaterinburg, Russia. [7] Ural Federal University, 620002 Ekaterinburg, Russia. Correspondence and requests for materials should be addressed to G.S. (email: sangiovanni@physik.uni-wuerzburg.de).

Iron and nickel are two of the most well-known ferromagnets, that is, conducting materials with a permanent magnetization[1]. Their importance comes primarily from the invaluable technological uses, ranging from invars with low thermal expansion, permalloys having high permeability, to maraging steel, high-resistance nichrome and corrosion-resistant coatings. Iron is also a cardinal ingredient of Earth's magnetism and its transport properties at high pressure are presently the object of a lively debate[2–6]. At first sight, nickel should not play a role in generating the Earth's magnetic field as it originates in the outer core, which is made of liquid iron. In current models, however there seems to be not enough energy to sustain the geodynamo through heat convection. Therefore, the importance of the inner core, ~20% of which is believed to be made of nickel, is being critically reconsidered[7].

Surprisingly, we still lack a complete theoretical comprehension of these two textbook materials. The reason can be ascribed to the intrinsic quantum many-body nature of their electronic structure, which makes a standard treatment in terms of independent electrons and conventional band theory inapplicable. The calculated Coulomb interaction is large and comparable in size in iron and nickel, which instead differ in the number of $3d$ electrons filling the bands close to the Fermi level. Iron is not too far from half filling, where the Coulomb interaction has the strongest effect and can easily drive a system Mott insulating. On the contrary, nickel has an almost full shell, a situation in which the Landau theory of Fermi liquids is in general recovered, even if the Coulomb interaction is significant. Yet, nickel was originally considered the more correlated of the two, because of photoemission satellites far away from the Fermi level, believed to originate from spectral weight transfer due to the Coulomb interaction[8,9]. A theoretical study by one of us[10] put the two materials on a similar level, stressing the existence in both of them of well-formed local moments, despite their marked itinerant character.

Here we go a step further and perform electronic structure calculations including the full local Coulomb interaction. This way we demonstrate that nickel would not be a strong-coupling quantum magnet without the van Hove singularities of its fcc density of states (DOS). In fact, it turns out that only the combined influence of its peculiar DOS[11] and of the electron–electron interaction can explain the Curie behaviour of its local spin susceptibility. This reflects in what we call pre-localized moments and a scattering rate, which is unexpectedly linear in temperature. The most important implication of our results for nickel comes from the observation that even at a pressure of hundreds of GPa, the position and the shape of these sharp features in the DOS do not change dramatically. Nickel remains in its fcc structure up to even larger pressures[12–14] and its magnetic moments, though smaller, are much more robust than those of iron[15]. Consequently, this physics is still active at the pressures of the inner Earth's core, at which instead iron is already a perfect Fermi liquid[5,6]. The non-Fermi-liquid mechanism identified here hence calls for including nickel in the current models of geomagnetism.

## Results

**Ferromagnetic transition temperatures.** The Curie temperature ($T_C$) of iron and nickel has been the object of several studies, in particular using the merger of density functional theory and dynamical mean-field theory (DFT + DMFT)[17–19]. This theoretical approach gives reliable results for three-dimensional materials with large coordination number, and is able to access the magnetic as well as the non-ordered phase above $T_C$. The latter is described by DFT + DMFT as non-vanishing local

magnetic moments with strong quantum fluctuations, which is crucial for the physics of correlated itinerant magnets[20–23]. Our DFT + DMFT spectra are consistent with previous calculations of similar kind[10,24–26] and agree reasonably well with angular-resolved photoemission data (see Supplementary Note 5 and Supplementary Figs 2–5). They also reproduce the known signatures of correlation in both materials, in particular the visible spin-polarized photoemission satellite around −6 eV for nickel. Single-site DMFT describes dynamical local correlations via a $k$-independent self-energy. This gives rise to the non-Fermi-liquid physics discovered in this paper. Yet, non-local exchange can also play an important role here and have a visible impact on the spectral function. It can be accounted for by quasiparticle self-consistent GW (QSGW)[27] and leads to a $k$-dependence of the self-energy. QSGW gives good results for iron but it does not fully capture the band structure of nickel[28]. A combination of both is possible in the GW + DMFT[29] or QSGW + DMFT method[28,30], but for the present extensive analysis such calculations are numerically too costly.

The hitherto published results differ in the Coulomb matrix elements $U_{ijkl}$ and in the approximation in which they are treated in the many-body part of the algorithm (DMFT)[10,24,31]. To calculate $T_C$, we use $ab$-$initio$-estimated $U_{ijkl}$ and go beyond the 'density–density' and 'Kanamori' parametrizations (see 'Methods' section, Supplementary Note 1, Supplementary Table 1, as well as ref. 32). This allows us not only to improve the agreement with the experimental transition temperatures but, even more importantly, to take a step forward in the understanding of the differences between these two itinerant magnets. In Fig. 1a,b, we show the DFT + DMFT magnetization curves. The agreement with the experimental $T_C$s is good (for the overestimate of $T_C$ in iron, see Supplementary Note 2 and Supplementary Table 2, as well as ref. 33). There is however, a substantial dependence on the parametrization of the Coulomb interaction, which reflects the big influence of electron–electron interaction in iron and nickel. However, since the two materials have different fillings of the $d$-shell and different band structures, the impact of correlations on their physics is profoundly contrasting. In Fig. 1c,d, we show the local self-energy $\Sigma(iv_n)$, a complex function describing the correction to the non-interacting propagation of an electron added to the system (Green's function) due to electron–electron interactions. $\Sigma(iv_n)$ highlights already many of the dissimilarities: Unlike in nickel, the $e_g$ orbitals in iron display strong scattering (proportional to the imaginary part of the self-energy at small frequencies) and large quasiparticle renormalization[34]. Notwithstanding the smaller scattering rate in nickel (Fig. 1d), we reveal a surprisingly large deviation from the $T^2$-behaviour predicted by the Landau theory of Fermi liquids. The scattering rate of nickel is indeed strongly non-quadratic in a large range of temperatures, as we will see in Fig. 4c.

**Iron as a typical strong-coupling ferromagnet.** The onset of ferromagnetic long-range order is signalled by a divergence of the $\mathbf{Q} = 0$ spin susceptibility at the Curie temperature. For a ferromagnetic metal, such as nickel or iron, the local spin susceptibility is instead a regular function of $T$. The latter is defined as $\chi_{loc}^{\omega=0} = \int_0^\beta d\tau \chi_{loc}(\tau)$, that is, the $\omega = 0$-Fourier component of the spin–spin response function $\chi_{loc}(\tau) = g^2 \sum_{ij} \langle S_z^i(\tau) S_z^j(0) \rangle$, where $S_z^i(\tau)$ is the local spin operator for the orbital $i$ on nickel or iron, at the imaginary time $\tau$. $g$ denotes the electron spin gyromagnetic factor. By studying $\chi_{loc}^{\omega=0}(T)$, the formation of local magnetic moments can be inferred. A Stoner-like ferromagnet does not have pre-formed local moments, whereas a strong-coupling ferromagnetic instability can be pictured as the emergence of localized spin

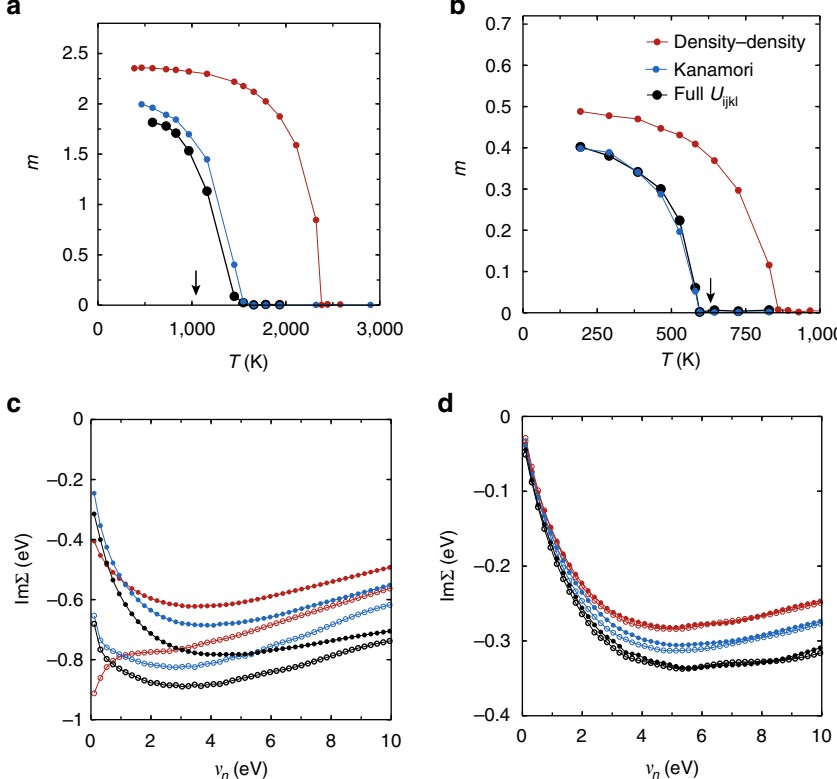

**Figure 1 | Curie temperatures and self-energies of iron and nickel.** Ferromagnetic order parameter $m = N_\uparrow - N_\downarrow$ in bcc iron (**a**) and in fcc nickel (**b**) as a function of temperature, whereas $N_\uparrow$ ($N_\downarrow$) are the total number of spin-up (-down) electrons in the correlated $d$ orbitals. The Curie temperature $T_C$ is signalled by the magnetization dropping to zero. For nickel, we obtain $T_C = 600$ K, very close to the experimental value of 633 K (ref. 16), indicated by the black arrow in **b**. Our estimated $T_C$ for iron is around 1,500 K, that is, roughly 30% larger than the experimental one of 1,043 K (ref. 16), marked by the black arrow in **a** (for a discussion, see 'Methods' section, where also a description of the three parametrization used for the Coulomb interaction is given). Imaginary part of the Matsubara self-energies of iron (**c**) and nickel (**d**) at $\beta = 30$ eV$^{-1}$ in the paramagnetic phase. The curves with filled circles show one of the degenerate $t_{2g}$ orbitals, with empty circles one of the degenerate $e_g$ orbitals. The lifetime of the quasiparticles is inversely proportional to Im$\Sigma(iv_n \rightarrow 0)$. Contrary to nickel, the scattering rate in iron at ambient pressure is large, even though the insulating-like shape of the density–density $e_g$-self-energy is replaced by an upturn at small frequencies in Kanamori and full Coulomb.

moments at high temperatures, which acquire phase coherence and order throughout the crystal on cooling. With DFT + DMFT, we can access $\chi_{loc}^{\omega=0}(T)$ also below the Curie temperature, as if magnetism was not present (paramagnetic solution). This is useful to study the intrinsic local spin response, independently of the actual long-range order. If the local moments are created by the electron–electron interaction, $\chi_{loc}^{\omega=0}(T)$ must behave, at sufficiently high temperatures, as $1/T$ (Curie law). This clearly happens in iron (Fig. 2a) for both parametrizations of the Coulomb interaction used in this case. Plotted on the same scale of the interacting one, the $U_{ijkl} = 0$-susceptibility (green dots), as well as the dressed convolution of two DMFT Green's functions ('bubble' approximation, grey squares), are small and weakly temperature dependent. They are completely overwhelmed by the pronounced Curie behaviour obtained, as soon as the Coulomb interaction is considered.

In the paramagnetic phase, iron is therefore a bad metal with a strong electron–electron scattering and robust local magnetic moments. The local susceptibility nicely fits to Wilson's formula[35]

$$\chi_{loc}^{\omega=0}(T) = \frac{\mu_{eff}^2}{3(T + 2T_K)}, \quad (1)$$

as shown in Fig. 2a,b. Here $\mu_{eff}$ is the local moment and, indeed, the estimated values are in excellent agreement with the experimentally ordered ferromagnetic moment. $T_K$ is the

Kondo temperature and indicates the screening of the local moment, as well as the onset of a Fermi liquid behaviour. As our results show, this would only occur far below $T_C$, if no long-range order would set in. At $T_C$, the paramagnetic phase of iron is therefore still closer to a local moment than to an itinerant electron description. This indicates that electronic correlations are much more important around $T_C$ than for the low temperature spin-polarized ferromagnetic solution, which is not too far from a single-electron Slater determinant.

**Nickel as a van Hove magnet.** The situation in nickel is shown in Fig. 3 and it is totally different. The absolute value of the local spin susceptibility (Fig. 3a) is way smaller than in iron. Also for nickel, it follows the Wilson law and, surprisingly, this is already the case for the non-interacting susceptibility. We can explain this by the particular shape of the DOS of nickel with its van Hove singularity, appearing in the $t_{2g}$ sector (see Fig. 3d) slightly below the Fermi level ($E_F = 0$). As shown in Fig. 3c, the corresponding band forms a hollow on the hexagonal face of the Brillouin zone, inside which the dispersion is essentially flat, except for a slow modulation between the six minima and the shallow maximum around the L point. Though this singularity is integrable in three dimensions, it induces a pre-localization effect of the Bloch electrons which, for geometrical reasons, have a band velocity $\mathbf{v} = 1/\hbar \nabla_{\mathbf{k}} \varepsilon(\mathbf{k})$ close to zero. This reflects directly in an apparent

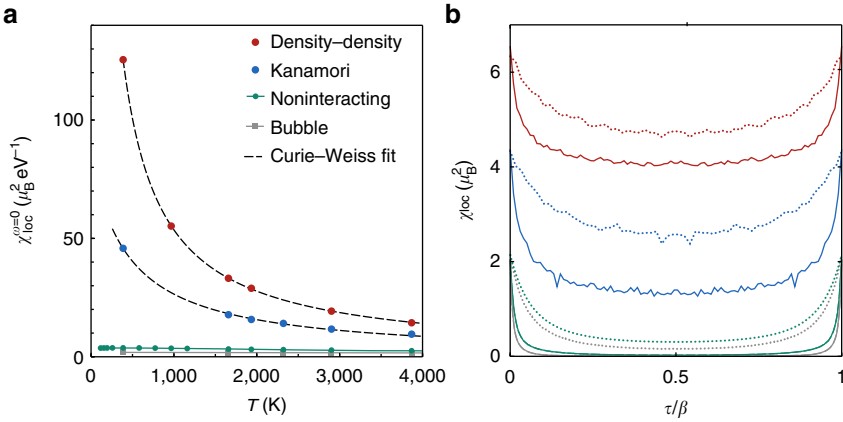

**Figure 2 | Paramagnetic spin response of iron at ambient pressure. (a)** Temperature dependence of the local spin susceptibility $\chi_{\mathrm{loc}}^{\omega=0}(T) = g^2 \int_0^\beta d\tau \sum_{ij} \langle S_z^i(\tau) S_z^j(0)\rangle$ in iron, calculated with DFT + DMFT. Calculations are performed following the non-ordered magnetic solution, also below the ferromagnetic transition temperature $T_C$ at which the uniform ($\mathbf{Q}=0$) susceptibility diverges (not shown). $\chi_{\mathrm{loc}}^{\omega=0}(T)$ in iron displays a marked $1/T$ Curie–Weiss behaviour, above as well as below $T_C$, indicating the existence of robust local magnetic moments. The fits are least-square fits with fit function $\chi_{\mathrm{loc}}^{\omega=0}(T) = \frac{\mu_{\mathrm{eff}}^2}{3(T + 2T_K)}$. For density–density, we obtain an effective local moment of $\mu_{\mathrm{eff}} = 3.97\mu_B$ and a Kondo temperature of $T_K = 35\,K$, for Kanamori $\mu_{\mathrm{eff}} = 3.14\mu_B$ and $T_K = 227\,K$. The DFT + DMFT data are compared to the uncorrelated and 'bubble' susceptibilities, that is, calculated, respectively, from the bare and 'dressed' Green's functions, neglecting the effect of vertex corrections. Figure **b** shows the decay in imaginary time $\tau$ of the local spin susceptibility at two temperatures $\beta = 4\,\mathrm{eV}^{-1}$ (dashed lines) and $\beta = 30\,\mathrm{eV}^{-1}$ (full lines), for density–density (red) as well as Kanamori (blue). The fact that $\chi_{\mathrm{loc}}(\tau = \beta/2)$ is going to zero much more slowly than $\beta$ implies the presence of persistent local moments at these temperatures.

Curie law of the non-interacting spin susceptibility, governed by the distance in energy from the flat band[11,36]. Including electronic correlations, the local susceptibility in Fig. 3b keeps its $1/T$ behaviour but is strongly enhanced (by a factor of 2–3). We stress that such enhancement is much stronger than what is to be expected from the quasiparticle renormalization $Z \sim 0.8$ ($Z = (1 - \alpha)^{-1}$ with $\alpha = \partial \mathrm{Im}\Sigma(iv_n)/\partial(iv_n)|_{v_n \to 0}$ in Fig. 1d). This indicates the fundamental importance of vertex corrections to the susceptibility of nickel, despite its large filling.

The second fingerprint of pronounced band effects in nickel is the kink in the spin susceptibility. This is present in the $U_{ijkl} = 0$ as well as in the interacting case (see Fig. 3a,b, Supplementary Note 10 and Supplementary Fig. 9) and separates two Wilson-like behaviours with different slopes of $1/\chi_{\mathrm{loc}}^{\omega=0}(T)$. The kink can be traced back to the fact that the chemical potential is $\sim 0.17\,\mathrm{eV}$ below the sharp upper edge of the $t_{2g}$ DOS and $\sim 0.05\,\mathrm{eV}$ above the van Hove singularity (see Fig. 3c,d). The former acts as a source of pre-localization too (for a demonstration, see the analysis under pressure and the inset to Fig. 4b). Since the two singularities are different, their influence to $\chi_{\mathrm{loc}}^{\omega=0}(T)$ is not symmetric, resulting in two different $1/T$ behaviours: decreasing $T$ from high temperatures, we switch from one to the other when the larger of the two energy scales, that is, $\sim 1,700\,\mathrm{K}$, is crossed (see Fig. 4a).

It is important to note that the inclusion of interactions does not alter this picture, apart from lifetime effects and broadening, which make the kink much smoother and its position slightly renormalized. The kink in the local susceptibility directly translates via the Bethe–Salpeter equation to a kink in the ferromagnetic susceptibility. It clarifies the experimentally observed kink at $\sim 1,200\,\mathrm{K}$ (ref. 37), reported almost 80 years ago but so far, to the best of our knowledge, without explanation.

Since in nickel, the Wilson-like behaviour is already present in $\chi_{\mathrm{loc}}^{\omega=0}(T)$ at $U_{ijkl} = 0$, a direct interpretation in terms of a Kondo effect is difficult. Let us hence turn to the imaginary time $\tau$ dependence of the local susceptibility, shown in Fig. 3e,f. Without interaction, there is a rapid drop of $\chi_{\mathrm{loc}}(\tau)$ for $\tau \to \beta/2$. This indicates the absence of long-lived local moments. A drop is

present with interaction as well, but on much larger timescales and from larger $\chi_{\mathrm{loc}}(\tau = 0)$ values. This confirms the presence of correlation-enhanced[38,39] local moments in nickel and the $\tau$ dependence reveals their screening properties. Since there remains a finite unscreened moment $\chi_{\mathrm{loc}}(\tau = \beta/2)$ even at $\beta = 30\,\mathrm{eV}^{-1}$ (370 K), the screening is not complete. Only at much lower temperatures (of the order of 100 K), this moment eventually gets completely screened, see inset of Fig. 3f. The existence of such a two-stage screening can be clarified invoking the few charge carriers available for screening, as in Nozières exhaustion scenario[40,41]. Also the local spin being larger than 1/2 (ref. 42) and the fact that $d$ electrons act both as localized moments and itinerant ones can contribute. The latter screen the former. This leads to different energy scales for the onset of screening and for complete screening[43], with the range in-between strongly affected by the van Hove peak in the $d$ electron spectral function.

**A unified picture.** A van Hove singularity is actually also present in the bcc DOS of iron (see Supplementary Note 8 and Supplementary Fig. 7). The reason why this does not influence the physical properties as for nickel is the different occupation of the $d$ shell of the two materials. Due to the proximity to half filling, in iron the physics is dominated by correlation-induced band renormalization and strong electron–electron scattering. This means that, even though the van Hove singularity does give a $1/T$ behaviour for the non-interacting $\chi_{\mathrm{loc}}^{\omega=0}(T)$ as well as a kink visible on a small scale (see Supplementary Fig. 7), this has hardly any influence in the actual behaviour in the presence of the interaction.

Nickel is instead close to nine $d$ electrons and moreover the individual occupation of its five $d$ orbitals is almost maximized. Hence, the entire manifold of electronic states is practically occupied and the majority of the active electrons lives close to van Hove singularity and to the sharp feature at the top of the $t_{2g}$ DOS. As a result, its physics is distinct from the more common case of a strongly correlated metal with sharp features in the DOS located far away from the band edges (bcc iron).

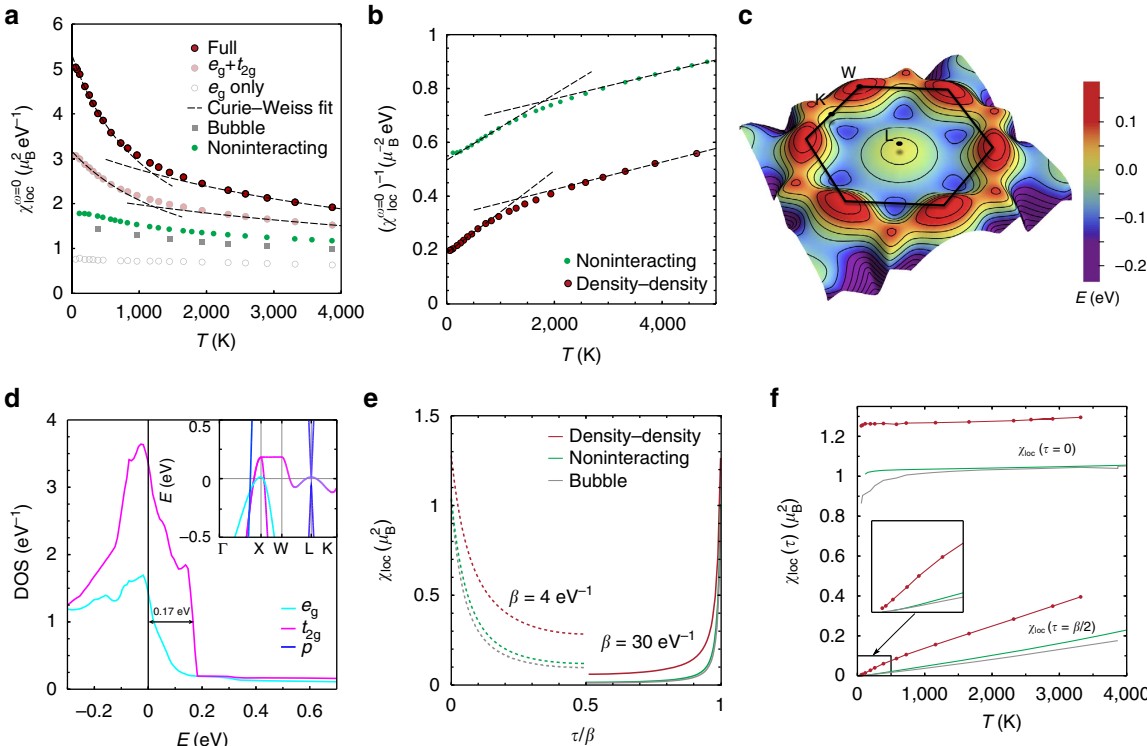

**Figure 3 | Paramagnetic spin response of nickel at ambient pressure.** (**a**) Temperature dependence of the static local spin susceptibility $\chi_{loc}^{\omega=0}(T)$ of nickel in DFT + DMFT, compared to the 'bubble' approximation and to the non-interacting one. To show that the characteristic temperature dependence of the non-interacting susceptibility originates from the $t_{2g}$ sector and its van Hove singularity, we also plot the intra-orbital contribution to the full $\chi_{loc}^{\omega=0}(T)$ ('full'), considering the cases where either the five orbital-diagonal terms of $\chi_{loc}$ are summed ('$e_g + t_{2g}$'), or only the two $e_g$ terms are retained ('$e_g$ only'). This shows the more conventional Pauli spin response of the $e_g$ part is in agreement with the fact that the $e_g$ DOS is much smoother around $E_F = 0$. (**b**) Inverse susceptibility for the non-interacting and DFT + DMFT case. This illustrates the main peculiarity of nickel, namely that already the non-interacting spin response is characterized by a $1/T$ law. As explained in the text, this is due to pre-localized moments arising from the vicinity to the van Hove singularity. (**c**) Electronic band dispersion of nickel on the hexagonal face of the Brillouin zone, close to the L point. The colors indicate the energy of the band relative to the Fermi level, which is located at zero energy. The extended flat region around the shallow maximum at L is responsible for the van Hove singularity. (**d**) $t_{2g}$ and $e_g$ DOS for energies close to $E_F = 0$. In the inset, the electronic state dispersion of nickel, close to the W-L-K region and to the top of the band is shown. The distance of the sharp step in the $t_{2g}$ orbitals at $E = 0.17$ eV corresponds to the kink of the non-interacting susceptibility in **b** at $T = 2,000$ K. (**e**) $\chi_{loc}(\tau)$ for $\beta = 4$ eV$^{-1}$ (dashed) and 30 eV$^{-1}$ (solid). (**f**) Instantaneous ($\tau = 0$) and long-time ($\tau = \beta/2$) values of $\chi_{loc}(\tau)$. From the latter one can clearly see that the moment is eventually screened at temperatures much lower than $T_C$. The comparison with the non-interacting and with the 'bubble' results shows also that vertex corrections are important and the DFT + DMFT result cannot be obtained by using 'dressed' quasiparticle propagators (see also **b**).

This difference between iron and nickel does not only concern the formation of local moments but it also leads to different microscopic mechanisms for ferromagnetism. In the case of iron, ferromagnetism is driven by Hund's rule coupling, which forms a large local moment of the $d$ electrons that are close to half filling. These Hund's moments then only need a bit of hopping to order[44]. For nickel on the other side, its peculiar DOS is essential not only for the formation of the local moments, but also for their ferromagnetic long-range order. This is the flat band[45], or more generally asymmetric DOS route[46,47], to ferromagnetism. In contrast to the aforementioned work, Hund's coupling plays also an important role in nickel. It enhances the local moments and strongly affects $T_C$ (see Supplementary Note 2 and Supplementary Table 2). Such a mixture of Hund's rule and flat band ferromagnetism has recently also been observed in BaRuO₃ (ref. 48).

**Nickel at Earth's core pressures.** Inspired by our improved understanding of the different nature of magnetism in iron and nickel, we discuss now the physics of Ni under high pressures. Indeed, if the non-Fermi-liquid properties of nickel survive the

extreme conditions of the Earth's core, our conclusions would be also important to the recent debate about geomagnetism. The key observation is that the van Hove singularity in nickel does not dramatically change its position with pressure. As shown in Fig. 4a, it gets gradually smoothed out but it survives pressures of even hundreds of GPa (see Supplementary Fig. 1, Supplementary Tables 4 and 5, as well as Supplementary Note 4 for the equations of state used to obtain the pressure–volume relationships for fcc Ni). At the same time, the sharp edge of the $t_{2g}$ DOS moves away from the Fermi level but its influence can be clearly observed in the non-interacting $\chi_{loc}^{\omega=0}(T)$ at all values of the pressure considered here. In Fig. 4b, the kink is indeed still visible at high pressure. In the inset, we compare the kink position with the distance—converted in temperature—between $E_F$ and the edge of the $t_{2g}$ DOS (see Fig. 4a). They scale the same way with pressure, confirming our interpretation in terms of sharp features of the DOS.

We now analyse the electron–electron scattering rate $\Gamma = -Z\text{Im}\Sigma(iv_n \rightarrow 0)$, as possible non-quadratic temperature dependencies of this quantity signal deviations from the Landau theory description of Fermi liquids. $\Gamma$ of pure fcc nickel is shown in Fig. 4c as a function of temperature at ambient pressure and

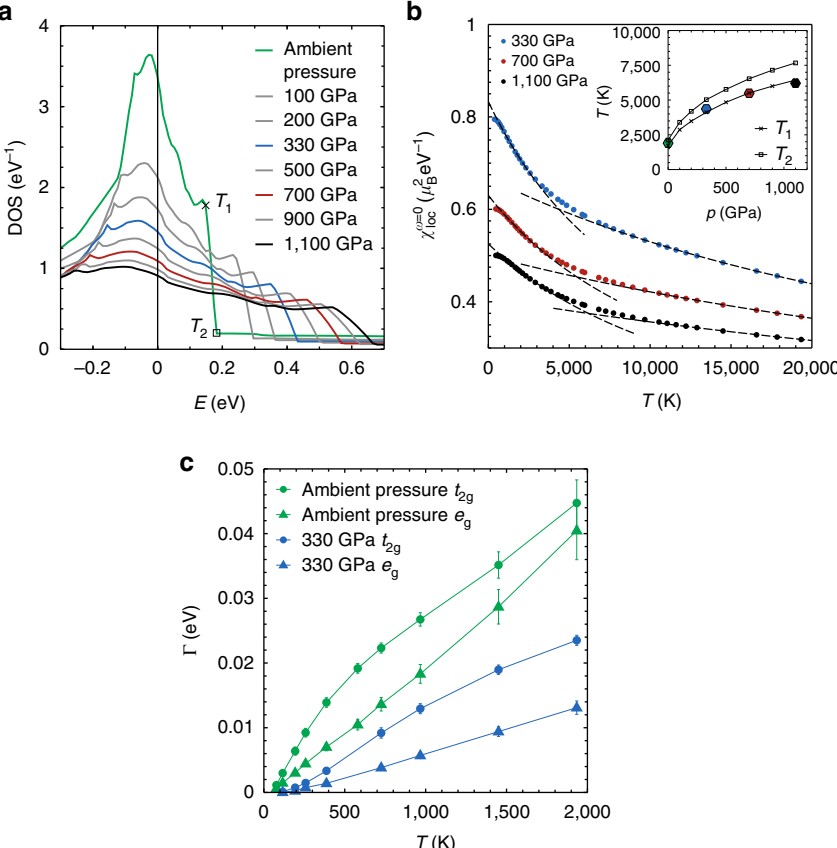

**Figure 4 | Nickel under pressure. (a)** The $t_{2g}$ DOS of nickel at different values of the pressure. The van Hove peak and the sharp edge are smoothed with increasing pressure, but the Fermi level ($E_F = 0$) remains close to these two singular points. The cross at temperature $T_1$ marks the beginning of the DOS steps, whereas the square at $T_2$ marks is its ending. Their evolution with pressure is shown in the inset to **b**. **(b)** Non-interacting local spin susceptibility versus temperature at high pressures. The dashed lines are least-square fits of the high- and low-temperature region, which uncover the separation between the two Wilson-like behaviours, shifting towards higher temperatures compared to ambient pressure. The position of this kink (shown as hexagons in the inset) scales with the energy distance between $E_F$ and the sharp edge of the $t_{2g}$ DOS. **(c)** A consequence of the sharp features at the top of the band is the deviation of the quasiparticle scattering rate from the $T^2$ law predicted by the Landau theory. Here this is shown for pure nickel, by plotting the scattering rates of one of the degenerate $t_{2g}$ and one of the degenerate $e_g$ orbitals, both at ambient pressure and at 330 GPa. The error bars come from averaging over different fit parameters (number of Matsubara frequencies and order of fit-polynomial).

330 GPa, characteristic of the Earth's inner core. In both cases, we observe non-Fermi-liquid behaviour in a large interval of temperatures. A technical comment is in order here: extracting the scattering rate from the Matsubara axis at these elevated temperatures poses well-known difficulties[6]. We therefore first checked that a three-orbital model with the same filling of nickel, the same bandwidth of the ambient pressure case and no singularity and no asymmetry in the DOS gives a nicely $T^2$ Fermi-liquid scattering rate (not shown). This confirms that it is possible to reliably extract the scattering rate from our DFT + DMFT calculations—at least for temperatures up to 2–3,000 K—without artefacts (see Supplementary Note 9 and Supplementary Fig. 8). As a matter of fact, even by considering error bars estimated from several polynomial fits to $Im\Sigma(iv_n \rightarrow 0)$ (denoted by the vertical bars in Fig. 4c), the non-quadratic temperature behaviour is clearly recognizable.

Hence we find that, even though judging from the self-energy nickel may seem to be a much more conventional metal than iron (see Fig. 1c,d), its scattering rate has a non-Fermi-liquid behaviour at ambient pressure as well as at 330 GPa. This deviation from the standard linear temperature dependence is caused by the van Hove singularity[36] and it is so robust that it is still visible at Earth's core pressures, where the

sharp features in the DOS are weakened but not yet completely gone.

**Disordered iron/nickel alloy.** The inner Earth's core contains about 20% nickel in addition to solid iron and, at these extreme pressure and temperature conditions, they form a disordered alloy. Therefore, to understand whether or not nickel should actually be considered in theories of geomagnetism, one has to study a disordered iron/nickel alloy under pressure. In particular, it should be seen if, even under extreme conditions, the electronic scattering rate is large and non-Fermi liquid, as in pure fcc Ni where the van Hove singularity mechanism is active. The electronic properties of an alloy are fundamentally different from those of a perfect crystal. To take the nontrivial disorder effects into account, we performed calculations for nickel alloys of various concentrations at the Earth's pressure within a coherent potential approximation (CPA) + DMFT scheme. The details of our implementation are described in the Supplementary Note 6.

The identification of the correct crystalline structure of iron at Earth's core conditions is still a subject of debate (see ref. 5 and references therein, as well as ref. 49). In this work we will consider an hcp Fe/Ni alloy, since hcp Fe (ε-Fe) emerges as the most stable

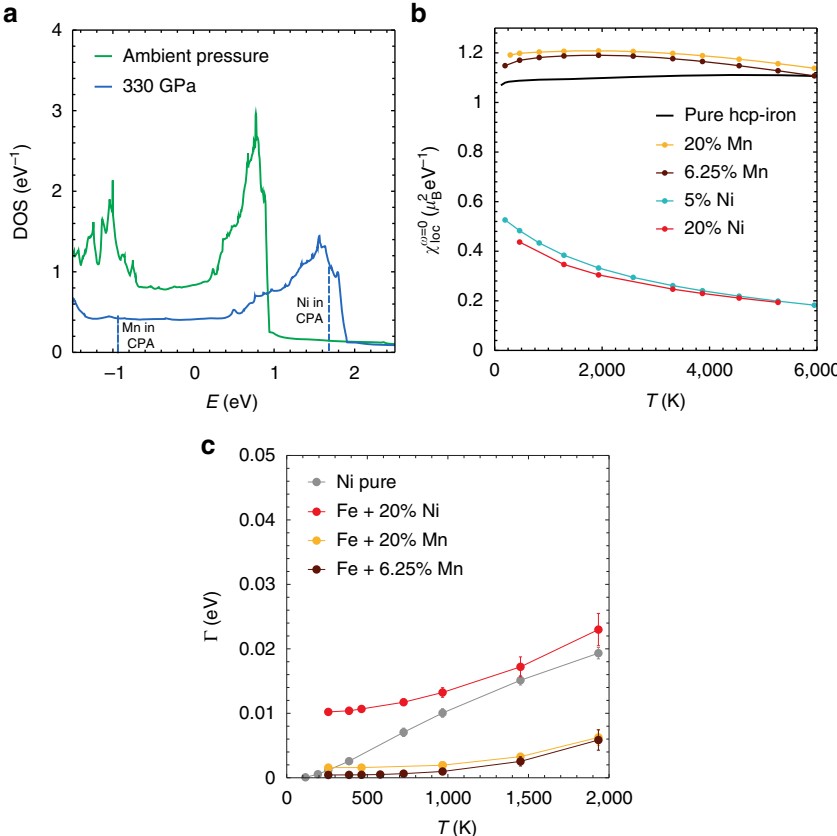

**Figure 5 | Iron/nickel alloy under pressure. (a)** DOS of the ε (hcp) phase of iron at ambient and Earth's core pressure (green and blue lines, respectively). $E = 0$ corresponds to the chemical potential of hcp iron. The vertical dashed lines indicate the energies at which the local levels of Ni and Mn are shifted within the CPA + DMFT scheme. In contrast to the shift for the Fe/Mn alloy (used here as a reference example), that for Fe/Ni brings the local levels of nickel close to the peak at the top of the hcp band. Since the latter is similar to the sharp features of pure fcc nickel (shown in Fig. 4a), the non-Fermi-liquid effects observed in the pure fcc material are expected to survive in the iron/nickel alloy. **(b)** Non-interacting local spin susceptibility for pure hcp iron and for the alloys of hcp iron with Mn and Ni, all at Earth's core pressure, that is, 330 GPa. The iron/nickel alloy strongly deviates from standard Pauli behaviour of good Fermi liquids. The example of Mn is on the contrary much closer to the Fermi-liquid result of hcp iron. **(c)** The scattering rate (averaged over all $d$ orbitals) of the nickel alloy at high temperatures within CPA + DMFT is remarkably similar to the corresponding one of pure fcc nickel at these pressures and much larger than that of the Fe/Mn alloy, taken here as a reference.

structure, according to several studies[14,50,51]. The blue curve in Fig. 5a shows the DOS of hcp iron at core pressure. The vertical dashed lines in the case under pressure show the position of the local levels, after the shift of the CPA + DMFT algorithm that simulates the presence of the dopant atoms. For the iron/nickel alloy, the shift is towards the top of the hcp band and brings the local levels of Ni close to the sharp structures resembling those of pure fcc nickel (shown in Fig. 4a). We also considered the alloy with manganese for reference purposes. In that case, the CPA shift has the opposite sign and it does not bring the energy levels close to any special structure in the DOS, therefore the alloy with Mn is expected to have more standard electronic properties than that with Ni.

The first indication that the hcp iron/nickel alloy shares part of the anomalous behaviour of fcc nickel comes from its non-interacting spin susceptibility (Fig. 5b). While pure hcp iron, as well as the alloy with Mn, displays a Pauli-like spin susceptibility in the absence of electron–electron interaction, the alloy with nickel (both with 5 and 20% nickel content) shows the Wilson-like behaviour which characterizes pure fcc nickel (see Fig. 4b). Note that the absolute values of $\chi_{loc}^{\omega=0}(T)$ depend mainly on the filling, as already observed for pure iron and nickel. Figure 5c confirms that, when taking the electron–electron interaction into account, 20% nickel leads to a linear and large

quasiparticle scattering rate at high temperatures. This confirms the close resemblance of the iron/nickel disordered alloy to pure fcc nickel, for what concerns the non-Fermi-liquid nature of the scattering. The red solid circles shown in Fig. 5c approach the grey curve at high temperatures (or even go somewhat above it), which refers to pure fcc nickel at 330 GPa. This demonstrates that the anomalous effect of nickel is relevant also in the case of the alloy at the Earth's core conditions.

## Discussion

The disorder, that we have treated here at the CPA level, turned out to have a crucial effect. A comparable amount of nickel combined with hcp iron in a translationally invariant crystalline structure gives more Fermi-liquid results, as interestingly pointed out in a recent paper by Vekilova *et al.*[52]. The latter approach has the advantage over CPA + DMFT that non-local changes to the band-structure induced by the presence of the nickel atoms are fully taken into account. By going beyond the state-of-the-art implementation of CPA + DMFT, which is the one we followed here, it will be possible in the future to fully describe this difficult interplay between many-body and disorder effects.

The linear in $T$ scattering rate of the iron/nickel alloy at core pressure in a large window of temperatures suggests a small

thermal conductivity. Molecular dynamics followed by real-space DMFT calculations of large nickel supercells, shows that the van Hove singularity in the spectrum and the scattering rate are not dramatically modified by the inclusion of thermal disorder (for details, see Supplementary Note 7 and Supplementary Fig. 6). Our results may therefore play a role in models of the geodynamo, recently questioned because of the large thermal conductivity of iron[2]. Whether or not this can be reconciled with iron alone is at the moment under debate[6,7], but according to the current understanding there would be too little energy left for convection in the total heat budget[4]. In light of our results, it will be interesting to reconsider the contribution of nickel to the total thermal conductivity of the core. The next challenge will thus be to include electronic correlations also in the *ab initio* study of the liquid phase of iron and nickel and determine if a more consistent explanation of the geodynamo can be obtained.

## Methods

**Density functional theory and projections.** Density functional calculations within the local-density approximation were performed using the projector-augmented-wave[53] based Vienna *ab initio* simulation package[54,55]. We used the experimentally determined lattice parameters for both materials, that is, 2.87 Å for iron and 3.52 Å for nickel. The local Hamiltonian for the subsequent many-body calculations was constructed using the projection on local orbitals as described in refs 56,57. Here we included the 3d bands as well as the 4s band of both metals.

**Pressure.** To simulate the effects of pressure in Ni, we calculated energy versus volume curves and fitted them with different equations of state (see Supplementary Note 4). From the fit, one obtains parameters, such as the bulk modulus, that can in turn be used to produce a pressure versus volume phase diagram for the material. From such a diagram, one can then read off the estimate the equation of state gives for the volume at any pressure. We employed different equations of state as well as different DFT functionals and created an aggregate best guess for the volume at each pressure considered. Since the equation of state of Fe is not well described within DFT, we used the volume established within the preliminary reference Earth model (see Supplementary Note 6 for futher details). Subsequently, a calculation using local-density approximation was performed using the chosen volume and a low energy model was constructed as described in the 'Density functional theory and projections' section.

**DFT + DMFT.** We combine DFT and DMFT using a localized basis constructed by means of the projection formalism as described in the 'Density Functional Theory and projections' section. From there we obtain a Hamiltonian matrix, which is subsequently exploited to compute the so-called hybridization function. This is used as an input for our hybridization-expansion continuous time quantum Monte Carlo (CT-HYB) impurity solver[58], which solves the auxiliary Anderson impurity problem within the DMFT loop numerically exactly at each step. Since our double-counting correction is self-consistently adjusted, we needed ∼50 DMFT iterations to reach convergence. The software used to produce the DFT + DMFT results is the CT-HYB Wien–Würzburg code package 'w2dynamics' (ref. 32), which is available on request to the corresponding author. The results for the disordered alloys at core conditions have been obtained within a CPA scheme + DMFT and thermal disordered has been simulated also by using molecular dynamics and DMFT. The corresponding implementations are discussed in Supplementary Notes 6 and 7.

**Parametrization of the Coulomb interaction.** We have estimated $U_{ijkl}$ in a fully *ab initio* way including the five 3d and the 4s orbital in the target space, by means of constrained-random phase approximation (cRPA)[59]. In 'w2dynamics' (ref. 32), these matrix elements have been used here with decreasing level of complexity: The 'full-Coulomb' parametrization considers the full $U_{ijkl}$ tensor and it is applied here for the first time to iron and nickel. For a recent discussion on the role of exchange, see also ref. 60. In the other two parametrizations, some entries are instead neglected, with a consequent reduction in the numerical effort. In the 'Kanamori'-only parametrization, the $U_{ijkl}$ elements compatible with full rotational invariance are left, whereas the corresponding spin–spin interaction is constrained to an Ising-like form in the case of the 'density–density', see Supplementary Note 1.

**Double counting.** The Coulomb matrix elements calculated within the cRPA contain naturally the cubic symmetry of the crystal. This means, that unlike in the usual spherically symmetric parametrization of the Coulomb matrix via Slater integrals $F^k$(ref. 61), the diagonal entries $U_{iiii}$ are different for the $t_{2g}$ and $e_g$ orbitals. This poses a challenge to the so-called double-counting correction of the DFT + DMFT formalism, which is usually estimated as an orbitally averaged

Coulomb interaction, see for example, ref. 62. Here we use the approach to constraint the total charge in the impurity, which is based on the Friedel sum rule[63]. We thus require the orbital occupancies of the interacting and non-interacting impurity Green's functions to coincide in self-consistency[64], see Supplementary Note 3 and Supplementary Table 3 for details.

**Data availability.** The data that support the findings of this study are available from the corresponding author on reasonable request.

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

## Acknowledgements

We thank Sergio Ciuchi, Domenico Di Sante and Olle Gunnarsson for comments and useful discussions. This work was supported by the DFG through FOR 1346 (A.H., A.L. and A.T. in its FWF subproject I-1395-N16), FOR 1162 (M.K.) and SFB 1170 'ToCoTronics' (G.S.). A.K. acknowledges the state assignment of FASO, Russian Federation (theme Electron, 01201463326) and Russian Foundation of Basic Research (project no. 17-02-00942) and K.H. the European Research Council under the European Union's Seventh Framework Program (FP/2007-2013)/ERC Grant agreement no. 306447. The authors gratefully acknowledge the Gauss Centre for Supercomputing e.V. (www.gauss-centre.eu) for funding this project by providing computing time on the GCS Supercomputer SuperMUC at Leibniz Supercomputing Centre (LRZ, www.lrz.de).

## Author contributions

A.H. and M.K. performed the *ab initio* many-body calculations, with the cRPA matrix elements by E.S. as input. G.S. conceived and coordinated the project. All authors have contributed to the physical interpretation of the results and worked together at the writing of the manuscript.

## Additional information

**Competing interests:** The authors declare no competing financial interests.

