## [Peer Review File · Nature Communications]

Reviewers' comments:

Reviewer #1 (Remarks to the Author):

The paper describes first principles calculations based on density functional theory (DFT) and dynamical mean field theory (DMFT) of the behaviour of iron and nickel, particularly in terms of local magnetism and the effect on the electron-electron (ee) scattering mechanism, which is relevant for the resistivity of the metal. The authors present convincing evidence that the ee scattering rate for nickel is linear in temperature, violating the conventional Landau theory of metals. The reason for this behaviour is tracked down to the presence of van-Hove singularities, which persist even at pressures as high as those found in the centre of the Earth. A consequence of this behaviour is that the resistivity of nickel is relatively high at the expected temperatures of the Earth's core. This peculiar behaviour of nickel is at variance with the normal Fermi liquid behaviour of iron, and so the authors propose that a presence of 20 % of nickel could be significant to increase the resistivity of the core.

The results are important and deserve to be published in some journal, however, I have one fundamental question for the authors, which I believe they need to address before the paper could be considered for Nature Communication.

It seems to me that all calculations have been performed on a perfect face-centred-cubic crystal. If this is the case, then there is a fundamental missing link with the Earth's core: thermal disorder. The whole theory is based on the presence of van-Hove singularities in the electronic density of states, but these singularities will very likely be destroyed by thermal disorder, both ionic disorder and thermal electronic excitations that at core temperatures cannot be neglected. Indeed, although local magnetism persists at Earth's core pressures and zero temperature, ionic disorder is enough to reduce it by approximately 50% and electronic excitations to 4000-6000 K are sufficient to quench it completely (at least at the DFT level).

The question then is to what extent, if to any at all, thermal disorder and electronic excitations are taken into account in the paper? The work is still valid and interesting, but if indeed thermal effects are important then I do not think that the paper is really relevant to the Earth's core.

One additional comment is that the authors should remove the reference to the Zhang et al. paper (their reference 6), as this paper was found to be wrong and has since been retracted.

Reviewer #2 (Remarks to the Author):

I enjoyed reading this manuscript. It is a nicely prepared work with interesting and important results and potentially important conclusions. Nevertheless, I cannot recommend publication in NC. It is hard to find any strong breakthrough that could motivate such a platform for disseminating the present findings. The conclusions regarding the Earth core composition and derived properties (heat convection) are not solid enough. Instead I suggest preparing a longer version and publishing in a specialized journal. My specific concerns are as follows.

1) The coupling between crystal structure and electronic structure properties is a well-known fact.

2) The coupling between correlation effects (DMFT level) and crystal lattice is also a well-documented issue (see e.g. Ref. 20).

3) The Fermi-liquid behavior of Fe at core conditions was discussed in Ref. 5. However, mostly the fcc and hcp structures were found to remain in Fermi-liquid state but not the bcc structure (which is believed to be the stable structure at core conditions).

4) Indeed the Earth core is supposed to contain up to 20% Ni. Whether this is segregated "pure" fcc Ni (actually a mixture of Fermi-liquid Fe and non-Fermi liquid Ni as suggested here) or a solid solution is quite unknown. The present work nicely demonstrates the interplay between two effects on pure Ni but its relevance to core materials is unclear.

Reviewer #3 (Remarks to the Author):

"Local magnetic moments in iron and nickel: Electronic correlation, van-Hove singularities and Earth's core pressure" by A. Hausoel and co-workers deals with electronic properties of iron and nickel at the advanced DFT+DMFT level and appears as a high-quality theoretical research. The paper is well written and offers novel and unexpected results. The physical mechanism described in the paper and the reported non-Fermi liquid behavior of nickel are of definite interest to scientists addressing strong electronic correlations in materials. It should also provide useful information for high-pressure researchers. No doubt that the results deserve publication in a timely fashion. What should make the paper suitable for a wide audience of Nature Communications is its relation to the physics of Earth's core and potential impact on the geophysics. It is underlined by the authors themselves and the concluding claim of the paper is that the importance of nickel in models of geomagnetism may be reconsidered. I think this extremely important point has to be clarified quite a bit. There is no doubt that iron alloy with nickel (around 10 at. % of the latter) is the main component of Earth's core. There is a tendency to ordering between Fe and Ni and what is normally expected at high-temperature conditions of Earth's core (> 5000 K) is a disordered alloy with possibly some degree of short-range order. As the authors discuss the unexpected behavior of pure fcc-Ni and its relevance to Earth's core, do they really mean that the phase separation between Fe and Ni is possible? They write, "the proximity to a hcp-fcc-liquid triple point may indicate that the inner core of the Earth could host a mixture of Fermi-liquid iron and non-Fermi liquid nickel" but do they read the phase diagram correctly? I think a thermodynamic consideration of the phase stability (order vs phase separation) is necessary. Further, there is a recent publication "Electronic correlations in Fe at Earth's inner core conditions: Effects of alloying with Ni" by Vekilova et al., Phys. Rev. B 91, 245116 (2015) considering a simplified model of Fe-Ni alloy at Earth's core conditions within DFT+DMFT. Their conclusion is different and the authors of the present paper should take that reference into account when formulating their conclusions.

Reply to Reviewer #1:

We thank the Reviewer for her/his positive and competent comments. In her/his report, the Reviewer describes the main message of our manuscript in a perfect way. This makes us confident that our text communicates the key ideas correctly.

It seems to me that all calculations have been performed on a perfect face-centred-cubic crystal. If this is the case, then there is a fundamental missing link with the Earth's core: thermal disorder. The whole theory is based on the presence of van-Hove singularities in the electronic density of states, but these singularities will very likely be destroyed by thermal disorder, both ionic disorder and thermal electronic excitations that at core temperatures cannot be neglected.

We fully agree with the Reviewer that thermal disorder plays a crucial role in this case. We thank her/him for this comment, which we addressed by performing Molecular Dynamics calculations combined with Dynamical Mean Field Theory (DMFT). The outcome of this demanding scheme is now shown in the new Fig. 5 of the revised Supplemental Information, where we also give details about the calculations. A full Molecular Dynamics + DMFT is a very challenging problem and it has not been implemented hitherto. Ours can be considered a first step in this direction: we indeed do not calculate forces with the full DFT+DMFT functional, but we generate a series of “snapshots” with standard DFT-based Molecular Dynamics and then we perform a DMFT calculation with the resulting cell containing several inequivalent atoms. The results of this calculation are reported in the Supplementary Information. They evidence that, for the parameters and the temperature we have chosen, the van-Hove singularity is not completely washed out by the thermal disorder. Even though more investigations are needed in this so far unexplored territory, we think that these new results strongly support the relevance of our conclusions for the physics of the Earth's core.

Besides the ionic disorder due to temperature, let us comment here also on the “intrinsic” many-body effect of temperature. Even in the case of a perfect crystal, the latter is fully taken into account by DMFT, in contrast to DFT. The DFT+DMFT spectra are in fact not only broadened by the Fermi function because the real and imaginary parts of the self-energy have an intrinsic and non-trivial temperature dependence, which reflects in visible changes to the electronic spectrum. In the figure below we illustrate this, in order to support this point of our Reply more transparently. While the temperature hardly changes the LSDA spectrum (left panels), it has a pretty strong effect on the DFT+DMFT spectral function (right panels).

We can therefore see how the smearing effect on the van-Hove singularity is stronger in DFT+DMFT than in LSDA. Our molecular-dynamics + DMFT scheme allows us however to conclude that the van-Hove is still

present, even though DMFT is performed on the non-perfect lattice. We thank the Reviewer for stimulating us towards the derivation of this nontrivial result.

The question then is to what extent, if to any at all, thermal disorder and electronic excitations are taken into account in the paper? The work is still valid and interesting, but if indeed thermal effects are important then I do not think that the paper is really relevant to the Earth's core.

As detailed in the answer to the previous point, our new calculations allowed us to clarify positively this crucial question raised by the Reviewer.

Let us additionally mention here that, stimulated by the questions of the other Reviewers, we have also performed Coherent Potential Approximation (CPA) + DMFT calculations in order to take into account the disordered nickel/iron alloy, relevant to the Earth's inner core.

One additional comment is that the authors should remove the reference to the Zhang et al. paper (their reference 6), as this paper was found to be wrong and has since been retracted.

We have removed this reference.

Reply to Reviewer #2:

We thank the Reviewer for finding our manuscript enjoyable to read, for her/his valuable comments and for considering our results interesting and important.

Stimulated by her/his criticism – shared also by Reviewer #3 (see below) – about the lack of an analysis for the iron-nickel alloy arguably present in the Earth's core, we have supplemented our code with a Coherent Potential Approximation (CPA) scheme to address precisely this issue. With this new tool we have run new calculations for different nickel concentrations in iron.

4) Indeed the Earth core is supposed to contain up to 20% Ni. Whether this is segregated “pure” fcc Ni (actually a mixture of Fermi-liquid Fe and non-Fermi liquid Ni as suggested here) or a solid solution is quite unknown. The present work nicely demonstrates the interplay between two effects on pure Ni but its relevance to core materials is unclear.

Our new calculation specifically addresses this point raised by the Reviewer.

The results are shown in the new Fig. 3 and they confirm that nickel increases the scattering rate in the alloy with hcp iron. The latter, without nickel, is a very good Fermi liquid. After the inclusion of nickel at the CPA level, the imaginary part of the self-energy becomes instead large at small frequencies and deviates from the temperature dependence predicted by the Landau theory (see new Fig. 3f).

We understand this crucial result by observing that (as shown in the new Fig. 3c) the shift of the local levels (which we estimate by making a supercell DFT calculation, as described in the Supplemental Information) in the case of nickel moves the chemical potential up in energy, close to the peak at the top of the iron-hcp density of states. The hcp lattice has indeed a shape of the density of states, which close to full filling resembles that of the fcc. As a result, the disorder-induced modifications to the physics are similar to those occurring in pure fcc-nickel.

In order to confirm that Ni is “special” in this sense, we have tried another alloy, the one with Mn (not because of its relevance for the Earth's core but just to demonstrate that not every CPA alloy gives a non Fermi-liquid scattering rate). The result of this comparison is striking: for Mn the shift is in the opposite direction and does not land in proximity to a sharp feature (see Fig. 3c) and indeed the alloy with Mn is a very good Fermi-liquid. At 2000 K the scattering rate of hcp-iron with 6.25% or 20% Mn content is about a factor of four smaller than with 20% Nickel. This is a very important result in light of the comments made by Reviewer #2 and #3.

We totally agree with the first two points raised by the Reviewer:

- 1) The coupling between crystal structure and electronic structure properties is a well-known fact.*
- 2) The coupling between correlation effects (DMFT level) and crystal lattice is also a well-documented issue (see e.g. Ref. 20).*

Including the electron-phonon interaction explicitly in this calculation would of course be interesting and would for sure have some quantitative effect, but it would not change our finding of a van-Hove induced non Fermi-liquid scattering rate. We are indeed discussing here a high-temperature effect, happening at temperatures much higher than the Debye temperature of fcc-nickel – which ranges between 500 and 1000 K going from ambient to 330 GPa pressure, see A. Campbell, *et al.* Earth and Planetary Science Lett., **286**, 556 (2009) and J. Pigott, *et al.* Geophys. Res. Lett. **42**, 10239 (2015).

Above the Debye temperature, the phonons can be safely considered as classical. Therefore, the electron-phonon contribution to the scattering rate, instead of going as T^3 , as below the Debye temperature, is linear in T . At very high temperature the scattering with phonons can be estimated to the second order to be roughly proportional to the mean square displacement of the phonon variable, which is proportional to the temperature. Assuming a constant electron-phonon matrix element and a typical phonon frequency of the order of the Debye energy one gets $1/\tau = \pi \cdot \lambda \cdot k_B \cdot T$. This would add to the electron-electron contribution to the scattering rate. Our DFT+DMFT calculation gives an electronic contribution to Γ of about 20meV at 1000K and larger at higher temperatures. Using the electron-phonon coupling λ for fcc-Ni reported in PRB **15**, 4221 (1977) the electron-phonon contribution to Γ is about 22 meV. This means that the “anomalous” (i.e. non Fermi-liquid) part coming from the electron-electron scattering that we find here is a sizable contribution to the total scattering rate and it

has a similar (linear in T) behaviour, as the electron-phonon one.

In iron the electron-phonon coupling is larger and since for hcp iron the electron-electron contribution to the scattering is very small (good Fermi liquid), people have mainly considered the electron-phonon scattering for estimating the resistivity of the Earth's core. Our results and this estimate using classical phonons show that if nickel is present, even if less abundant than iron, the electronic contribution to the scattering rate, the electrical resistivity and to the thermal conductivity must be considered.

We would also like to comment on the third point raised by the Reviewer:

3) The Fermi-liquid behavior of Fe at core conditions was discussed in Ref. 5. However, mostly the fcc and hcp structures were found to remain in Fermi-liquid state but not the bcc structure (which is believed to be the stable structure at core conditions).

The discussion mentioned by the Reviewer about the Fermi-liquid nature of iron, as well as about its structural properties under high pressures in the presence of small concentrations of other elements, is relevant and not yet fully settled in the literature. Our contribution here regards instead the effect of an alloy with nickel. Nickel is indeed, together with iron, the main constituent of the inner core of the Earth and we believe it is important to determine its effect starting from the most stable high-pressure phase of pure iron according to DFT calculations: the hcp one.

Thanks to our new CPA calculations we can clearly conclude that 20% of nickel is enough to give a strong non Fermi-liquid behaviour of hcp-iron. This result is hence relevant for the Earth's core.

Since our new calculations, in particular the ones with CPA for the nickel-iron alloy enrich our analysis and make our conclusions more solid, we are confident that Reviewer #2 can now suggest the manuscript for publication in *Nature Communications*.

Reply to Reviewer #3:

We thank the Reviewer for her/his competent comments and for finding ours a work of “*high-quality theoretical research*”, “*well written*” and that it “*offers novel and unexpected results*”.

The main criticism regards the lack in our first version of results for the alloy:

There is a tendency to ordering between Fe and Ni and what is normally expected at high-temperature conditions of Earth’s core (> 5000 K) is a disordered alloy with possibly some degree of short-range order. As the authors discuss the unexpected behavior of pure fcc-Ni and its relevance to Earth’s core, do they really mean that the phase separation between Fe and Ni is possible?

This point is absolutely well taken. To address it, we made a big effort and improved our quantum Monte Carlo DMFT code so that it can now treat the nickel-iron alloy within the Coherent Potential Approximation (CPA). With these changes we have then performed new calculations for different nickel concentrations. We establish that the non Fermi-liquid behaviour of the scattering rate detected in pure fcc-nickel still persists in the alloy 20% Ni / hcp-Fe. This allows us to make the contact between our results and the Earth’s core much closer.

They write, “the proximity to a hcp-fcc-liquid triple point may indicate that the inner core of the Earth could host a mixture of Fermi-liquid iron and non-Fermi liquid nickel” but do they read the phase diagram correctly?

We thank the Reviewer for pointing out the ambiguity of this sentence, in the absence of an analysis of the thermodynamic stability. We have removed the sentence after having done explicit calculations for the Fe/Ni alloy under pressure.

Further, there is a recent publication “Electronic correlations in Fe at Earth’s inner core conditions: Effects of alloying with Ni” by Vekilova et al., Phys. Rev. B 91, 245116 (2015) considering a simplified model of Fe-Ni alloy at Earth’s core conditions within DFT+DMFT. Their conclusion is different and the authors of the present paper should take that reference into account when formulating their conclusions.

We added this recent paper to our references, as it is extremely relevant for our study. We thank the Reviewer for drawing our attention to it.

In the new version we also comment on the differences between our CPA+DMFT and their approach.

In short, our CPA approach is in a sense complementary to that used there. Vekilova, *et al.* model the alloy through a translationally-invariant crystal with a Fe_3Ni supercell where the Ni atom is in one single specific position. On the other hand, our CPA approach explicitly treats the 20% nickel (randomly distributed) disorder. Here, let us also stress here that in our CPA+DMFT calculation there are two reasons for a non-zero imaginary part of the self-energy (beyond the intrinsic one from DMFT): the first is the shift mentioned above, which can bring the impurity close to a sharp peak in the DOS. The second is an intrinsic $1/\tau$ -imaginary part, originated by the disorder averaging. This would be present even in the cases where the energy shift is not, per se, a source of non Fermi-liquid behaviour.

Reviewers' comments:

Reviewer #1 (Remarks to the Author):

The authors have addressed the question of the potential modification of the spectral function by thermal disorder. This has been a significant undertaking, and I am satisfied with the evidence they present that in this particular case ionic thermal disorder does not seem to significantly affect the density of states.

I also see that they have addressed the points about configurational disorder, raised by the other referees. I will not specifically comment on those.

I feel able to recommend publication in Nature Communications.

Reviewer #2 (Remarks to the Author):

The authors made substantial efforts to remedy the shortcomings of this manuscript. In particular they carried out additional alloy calculations based on mean-field approximation and estimated the effect of thermal disorder. I appreciate these attempts which clearly enhance the potential impact of their work. However, there are several unclear points which were not or only marginally addressed. I should mention the effect of crystal structure. All alloy calculations consider hcp lattice whereas the present information on the Earth's core suggests other lattices such as the bcc structure (which could in principle explain the observed core anisotropy in contrast to the hcp lattice). This makes the connection between the present academic work and the Earth's core much weaker than suggested in the manuscript. I still believe that the presented work is of high quality and deserves publicity in an appropriate specialized journal.

Reviewer #3 (Remarks to the Author):

The referees addressed the important issues and urged the authors to clearly show the relevance of their novel theory to the Earth's core.

I think the authors did their best trying to answer all the questions raised by the referees. Accordingly the paper has been adjusted, improved and extended, perhaps beyond what would be typically expected.

I appreciate the new CPA part, now the text is better related to the Earth's core. Certainly, there are still open questions even in this part, but the definite leap forward is visible. Concerning the thermal disorder, that is certainly a very important point. However, I believe one should avoid formulating impracticable tasks for the authors. There is a possible effect of strong electron correlations in iron alloyed with nickel at extreme pressure/temperature conditions of the Earth's core. There is an obvious effect of thermal atomic vibrations. In an ideal world one would treat them simultaneously. However, there is currently no possibility to run any proper molecular dynamics (MD) simulation within DMFT. It is just not feasible with the available supercomputers. I am rather surprised with the brave attempt by the authors to go into MD. That deserves my respect. Though, certainly, a 27-atom supercell is ridiculously small (to be short, the phase space (atomic displacements + momenta) is not properly covered during such an MD run, so any averages are doubtful anyway). Accordingly the presented proof is rather vague (by the reason above and a couple of others, such as the averaging procedure itself). But it provides some indication, and again I do not think anyone can request the authors to do this task properly. To me the important finding is the revealed Ni effect. Yes, it may or may not survive at high temperature. But if it does, the implication might be really important to geophysics.

I also notice that not only one cannot be sure that the hcp structure of iron is thermodynamically stable at the Earth's core conditions; one can hardly claim that it is the most stable high-pressure phase of pure iron according to DFT calculations. As the free energy differences among hcp, fcc,

and bcc are rather tiny at the Earth's core conditions, the convergence of the MD simulations with respect to the simulation cell size is crucial. No converged results have yet been published. However, hcp is by all means one of the best candidates according to the published data. Perhaps, if the proper structure turns out to be, say, body-centered cubic, the effect of Ni could be different. That might be of interest for future studies.

Another important issue, which I have to address, is related to the recent discussions, initiated by the GW-community. The applicability of LDA+DMFT as a method to study the compressed iron or nickel has been questioned. I am afraid this issue cannot be left untouched in the present paper. The doubt is based on the fact that DMFT uses a very limited basis set (like five d-orbitals in the present case) and its self-energy is momentum independent. There are claims that under high pressure momentum dependence should increase, and that fact should be discussed. No one doubts that DMFT-approximation should be fine if the physics is really very localized (like 4f-states in rare earth materials), but there are indications that in Fe and Ni the physics is rather nonlocal even at normal conditions. The recent publications suggest that LDA itself does not correctly describe the spectrum of Fe at normal conditions (see Fig.1 in Arxiv:1603.05521 as an example). LDA+DMFT corrects the spectrum only partially (PRL 103, 267203(2009) and PRB 93, 205151 (2016)). First paper is LDA+DMFT, second is Gutzwiller DFT, which for Fe is basically the same. Particularly, there is a noticeable disagreement between the calculated and experimental mass enhancement (see the same references). As it follows from Table II (the second reference), for certain k-points LDA+Gutzwiller is even worse than GGA. This fact raises speculation that correlations, not included in local DMFT, could be of high importance for Fe. And indeed, there is some support for this point. As it is shown in Arxiv:1603.05521, the ARPES spectrum is reproduced perfectly well if one uses self-consistent quasiparticle GW method (QSGW). This method uses the same basis set as LDA (i.e. not only the d-orbitals on the same site as DMFT) and its self-energy depends on momentum. Here is the conclusion by the authors of Arxiv:1603.05521: "If self-energy is k-averaged to simulate a local self-energy, the QSGW band structure changes significantly and resembles the LDA. Thus non-locality in the self-energy is important in transition metals, and its absence explains why LDA+DMFT does not yield good agreement with ARPES". The discussion above indicates that there could be some problems with LDA+DMFT when it is applied to Fe at normal conditions. When material is compressed, momentum-dependent correlations might become progressively more important (collective type excitations), whereas local correlations (which DMFT considers) might become suppressed. That means that all-electron approaches with momentum dependent self-energy like QSGW might work better and better, whereas DMFT might unfortunately fail, if it fails even without pressure. I suppose that it is important that the authors, being experts in DMFT, give a plausible answer to this criticism by adding the corresponding discussion into the main text or the supplementary material.

To summarize, I think the paper may become suitable for publication in Nature Communications.

Reply to Reviewer #2:

We are glad that the Reviewer appreciates the improvements to our manuscript, after our new alloy calculations. She/He nevertheless still raises a further comment that we would like to address below.

All alloy calculations consider hcp lattice whereas the present information on the Earth's core suggests other lattices such as the bcc structure (which could in principle explain the observed core anisotropy in contrast to the hcp lattice). This makes the connection between the present academic work and the Earth's core much weaker than suggested in the manuscript.

The Referee's comment is pertinent to the subject, but we believe that it does not reduce the relevance of our findings. Our publication, on the contrary, would bring a significant and timely contribution to the forefront research on the Earth's core magnetism. It is true, there is presently a debate on the structural composition of the Earth's core. But, at the moment, it is not possible to conclude unambiguously whether the iron in the core crystallizes in a pure bcc phase or in a pure hcp or in a mixture of both. Let us also refer to Reviewer #3 who holds that *"hcp is by all means one of the best candidates"* in this respect. Moreover, in addition to ~10-20% of nickel, there are also small concentrations of other elements which complicate the analysis.

Yet, our result is important and clear: the impact of nickel on hcp-iron at this pressure and at this temperature is unexpectedly strong. The non-Fermi liquid properties that such an alloy displays should therefore be considered in future models for the geomagnetism.

As Reviewer #3 also writes that *"according to the published data"* hcp is the most probable structure for iron in the core. Should this be disproved, the results of our manuscript would still be of relevance: they would indeed force to investigate whether or not nickel induces a breakdown of the Fermi liquid scattering rate also in the new structural iron environment.

Reply to Reviewer #3:

We thank the Reviewer for finding our results an important step forward and for acknowledging the effort done after the first round of review. She/He carefully read our revised version and we are thankful for the competent and deep comments of her/his second report. Below we address all of them.

I also notice that not only one cannot be sure that the hcp structure of iron is thermodynamically stable at the Earth's core conditions; one can hardly claim that it is the most stable high-pressure phase of pure iron according to DFT calculations. As the free energy differences among hcp, fcc, and bcc are rather tiny at the Earth's core conditions, the convergence of the MD simulations with respect to the simulation cell size is crucial. No converged results have yet been published. However, hcp is by all means one of the best candidates according to the published data. Perhaps, if the proper structure turns out to be, say, body-centered cubic, the effect of Ni could be different. That might be of interest for future studies.

We fully agree with the Reviewer. These considerations also relate to the comments by Reviewer #2, to whom we have answered above. We are convinced that the inspiration for future studies is also an element in support of the publication of our work.

Another important issue, which I have to address, is related to the recent discussions, initiated by the GW-community. The applicability of LDA+DMFT as a method to study the compressed iron or nickel has been questioned. I am afraid this issue cannot be left untouched in the present paper. The doubt is based on the fact that DMFT uses a very limited basis set (like five d-orbitals in the present case) and its self-energy is momentum independent. There are claims that under high pressure momentum dependence should increase, and that fact should be discussed. No one doubts that DMFT-approximation should be fine if the physics is really very localized (like 4f-states in rare earth materials), but there are indications that in Fe and Ni the physics is rather nonlocal even at normal conditions [...]. I suppose that it is important that the authors, being experts in DMFT, give a plausible answer to this criticism by adding the corresponding discussion into the main text or the supplementary material.

The Reviewer touches upon several fundamental points here. We address them in detail below. It is certainly true that the QSGW method is superior to and also conceptually more appealing than DFT and its extensions in the form of DFT+X approaches. It captures some correlation physics correctly and, in particular, the exchange. In the cases where DFT+X is not sufficient on its own, GW provides ones with a greatly improved "single particle" starting point for methods like DMFT.

This does not automatically mean, however, that QSGW alone will yield better agreement with experiment than DFT+X approaches. Especially concerning transition metals and their oxides we would say that certainly QSGW is capable of describing NiO in the ordered AFM phase reasonably well (PRB 76, 165106), but this is not the case for correlated paramagnetic phases. In this respect QSGW or the GW approximation in general is similar to LDA or Hartree-Fock. This is for instance observed in NiO above the Neel temperature where DMFT is absolutely crucial (PRL 99, 156404). Of course, since DFT+DMFT "inherits" its k -dependence from the DFT starting point, if that starting point is bad the DMFT will have a hard time correcting it.

So, being clearly superior to DFT, QSGW augmented by dynamical local correlations could become the method of choice for spectra in the future, when the methodological details of the approach are ironed out (PRB 76, 165106). For certain quantities, like two-particle magnetic correlation functions there is, however, no way around DFT+DMFT for the time being, due to the computational complexity of these quantities. We comment on that in the new version of the manuscript and refer to the corresponding literature.

For the present cases, i.e. iron and nickel at normal conditions, we have performed additional calculations intended to compare k -resolved DFT+DMFT spectra with ARPES data. The results are shown the Figure below, which we have also added to the Supplementary information. The ARPES data shown for Ni are from of Himpsel *et al.* (PRB 19, 2919), while those for Fe are by Schäfer *et al.* (PRB 72, 155115). Here, we have used the results of the density-density interaction, since the QMC data at low temperature have the least noise.

For Ni the agreement between ARPES and DFT+DMFT is quite satisfactory close to the Fermi level along **L- Γ** , but is less good at large energies as well as along **Γ -X**. For QSGW (without DMFT augmentation) the situation is in fact a bit worse and, according to Sponza *et al.*, one should do QSGW+DMFT to get a good agreement for the whole band structure. From this one might conclude that non-local effects are important in ferromagnetic Ni.

For Fe the agreement between our data and ARPES is excellent along the **N- Γ -P** lines of the band structure. Larger quantitative discrepancies appear along **P-H- Γ** , which is a trend also seen in the Gutzwiller-DFT of PRB 93, 205151. Interestingly, **Γ -H** also poses difficulties, to a minor degree, for the QSGW of Sponza *et al.* in PRB95, 041112(R) (2017). A more quantitative analysis is of course in order, but based on the direct comparison with ARPES we would not say that LDA+DMFT fails to describe the ARPES spectrum of iron. For iron the interpretation of the comparison between ARPES and theory is less conclusive as one gets a good agreement both with DFT+DMFT and with QSGW without DMFT. In this case, only a simultaneous analysis of one- and two-particle quantities within the same theoretical scheme can clarify the relative role of local (DMFT) and non-local (QSGW) correlations.

Let us also note that the conclusion on the irrelevance of local correlations in iron by means of QSGW in Sponza *et al.* has been obtained by assuming a strongly reduced value of U for Fe (5eV), with respect to Ni (10eV), while our analysis based on cRPA gives similar (moderate) values for both compounds. Also, in the pioneering GW+DMFT paper by Biermann *et al.* (PRL 90, 086402) 3.2eV are employed for U . This discrepancy might be an effect of the larger basis set used by Sponza *et al.*, but nevertheless both points are worthy of further detailed investigation. It is not a point of controversy that a momentum dependent self-energy plays a role and this will certainly trigger future research in this direction, especially considering the current effort of the community to consistently improve the performance of GW+DMFT algorithms. Let us also note that GW+DMFT calculations as extensive as those of our paper (even including CPA) are not feasible with present day computational resources, and that a proper formalism to calculate two-particle correlation functions in GW+DMFT such as the studied spin-spin susceptibility still needs to be developed.

We agree with the Reviewer that pressure may change the balance between local and non-local correlations. In order to quantify this we have performed a preliminary G_0W_0 calculation with and without pressure. This can be seen in the figure below.

Comparison of LDA with $G_0W_0@LDA$ band structures at normal conditions (left) and at the volume of Ni at 330GPa as identified with our equation of state fits (right). In both panels the red curve is LDA, while the blue one is $G_0W_0@LDA$. We have adjusted the energy window to capture roughly the same 3d-band block of Ni close to E_F in both cases.

Our results appear to indicate that the k -dependent correction of the G_0W_0 -self-energy to the band structure gets smaller with pressure. This indicates that, in nickel the error made by taking DFT instead of GW as starting point for the DMFT becomes smaller and smaller, the larger is the pressure. Nevertheless, this does not contradict the general expectation of the Reviewer that the local correlations are reduced too, because the local self-energy of DMFT becomes also smaller upon increasing pressure due to of the higher itinerancy of the system. As there are claims as those mentioned by the Reviewer of an opposite trend of k -dependent correlation effects as a function of pressure and as GW can be made in several different flavours, our preliminary results cannot be considered conclusive. Since one of the main points of our manuscript is the magnetic susceptibility, which is too hard for GW+DMFT (see above), we prefer not to further elaborate on this in the manuscript.

There is also another aspect to be considered: the non-local correlations should be stronger in the ferromagnetic state than in the paramagnetic one, which most of our calculations are based on. Some evidences pointing in this direction are that all calculations for bcc iron show local moments above T_c , but the behaviour below T_c in DFT+DMFT looks Fermi-liquid like. The local quantities (electron damping, local susceptibility, etc.) that are discussed in our manuscript are relatively well captured by DFT+DMFT in the paramagnetic (bcc) phase. For the hcp phase the quality of DFT+DMFT results should be even better, since there are no long-range magnetic correlations.

To conclude, we agree with the Reviewer that a discussion on the limitations of DFT+DMFT in describing the spectrum of iron and nickel is necessary, as is a more detailed discussion of the complementary QSGW calculations. In the revised version of our manuscript we have dedicated a paragraph to that, adding the proper references. In the Supplementary information we report and discuss the k -resolved spectra shown in the present response. We thank the Reviewer for having stimulated us to make this further analysis, which definitely enriches our work and will likely trigger future forefront studies in this direction.

REVIEWERS' COMMENTS:

Reviewer #3 (Remarks to the Author):

I think the authors did a really good job in addressing the points raised by the referees. The amount of work and its quality is impressive. Yes, not everything became crystal clear, there are still points where we can argue, and certainly more things need to be taken into account. The authors have proven that they perfectly understand that. But they have done as much as it is currently possible to do and even more than that. The rest is too difficult or even unfeasible at the moment. So one should be realistic. Therefore I state that I am satisfied with the new version of the manuscript and I think it should be accepted and published as soon as possible. This paper deserves publication in Nature Communications.